# Temporal patterns and risk factors of diarrheal comorbidity among children aged < 5 years in rural western Kenya: Evidence from three consecutive enteric studies, 2008–2024

Billy Ogwel[1]*, Bryan O. Nyawanda[1,2,3], Brian O. Onyando[1], Alex O. Awuor[1], Caleb Okonji[1], Raphael O. Anyango[1], Caren Oreso[1], Catherine Sonye[1], John B. Ochieng[1], Stephen Munga[1], Dilruba Nasrin[4], Karen L. Kotloff[4], Patricia B. Pavlinac[5], Richard Omore[1ʘ], Elizabeth T. Rogawski McQuade[6ʘ]

1 Center for Global Health Research, Kenya Medical Research Institute- (KEMRI-CGHR), Kisumu, Kenya, 2 Swiss Tropical and Public Health Institute, Allschwil, Switzerland, 3 University of Basel, Basel, Switzerland, 4 Department of Medicine, Center for Vaccine Development and Global Health, University of Maryland School of Medicine, Baltimore, Maryland, United States of America, 5 Department of Global Health, University of Washington, Seattle, Washington, United States of America, 6 Department of Epidemiology, Emory University, Atlanta, Georgia, United States of America

ʘ These authors contributed equally to this work as joint senior authors.

* ogwelbill@gmail.com

## Abstract

Sub-Saharan Africa bears the highest burden of diarrhea, often complicated by comorbidities that delay diagnosis, hinder treatment, and worsen outcomes. As the epidemiology of diarrheal disease evolves, understanding comorbidity patterns is critical for effective public health responses. We examined the temporal patterns and risk factors of diarrheal comorbidity in Kenyan children aged < 5. We conducted secondary pooled analysis with a retrospective cohort design leveraging data from the Global Enteric Multicenter Study (GEMS, 2008–2012), the Vaccine Impact on Diarrhea in Africa (VIDA, 2015–2018), the Enteric for Global Health (EFGH) *Shigella* surveillance study (2022–2024). The outcome was comorbidity count, defined by Integrated Management of Childhood Illnesses case definitions and clinician diagnoses of ten conditions: malaria, bacterial infection, pneumonia, severe acute malnutrition (SAM), meningitis, acute febrile illness (AFI), respiratory Illness (non-pneumonia), anemia, stunting and wasting. Temporal trends were assessed using descriptive statistics and the Cochran-Armitage trend test. Risk factors were identified using generalized estimating equations with a Poisson distribution, adjusting for clustering. We analyzed data from 4,148 children with moderate-to-severe diarrhea; 90.3% had ≥ one comorbidity, with a declining trend across studies: GEMS (92.9%), VIDA (89.3%), and EFGH (86.6%). Pneumonia (49.5%), malaria (48.3%), and stunting (24.7%) were most common comorbidities. The proportion of children with only one comorbidity increased (28.9% [2008] to 49.7% [2024]), while multiple

**Data availability statement:** The GEMS and VIDA datasets are publicly available on ClinEpiDB (https://clinepidb.org/ce/app) while the EFGH data is publicly available through the Vivli data-sharing platform (https://search.vivli.org/doiLanding/studies/PR00011860/isLanding).

**Funding:** This work was funded by the Gates Foundation (grant INV-079156; RO, PP) through funding to the Enterics for Global Health (EFGH) Consortium as part of Phase C activities. The funders had no role in study design, data collection and analysis, decision to publish, or preparation of the manuscript.

**Competing interests:** The authors have declared that no competing interests exist.

comorbidities declined. Traditional comorbidities (malaria, pneumonia, wasting, SAM) significantly decreased, while AFI, anemia, and non-pneumonia respiratory illness increased. Multivariable analysis identified older age, lower caregiver education, dehydration, vomiting, 3-month lagged rainfall and temperature, and high respiratory rate as drivers of higher comorbidity counts, while female sex was associated with fewer comorbidities. Despite the high prevalence, we observed a 20–23% decline in comorbidity burden and a fundamental shift in disease profiles. Our findings support the need for a shift from single-disease control to integrated disease management.

## Introduction

While diarrheal deaths have decreased, diarrhea remains a major public health problem, disproportionately affecting sub-Saharan Africa [1]. This challenge is amplified by comorbidities, co-occurring conditions, which complicate the clinical course of an index disease [2]. The presence of comorbidities increases disease complexity and severity, hinders diagnosis, delays appropriate treatment, and elevates the risk of adverse outcomes and associated costs [3,4]. These effects might be mediated by factors such as increased physiological strain, resource competition, modification of the gut flora and strain on organ systems [3].

Previous research on diarrheal comorbidity in children has largely focused on disease pair comparisons with prevalent conditions such as acute respiratory illness [5], malaria [6] and anemia [7]. This approach, while valuable in understanding specific disease interactions, may not fully capture the complex landscape of multiple comorbidities often present in children with diarrhea. Furthermore, the landscape of childhood diarrheal illness has undergone significant transformations since 2008, driven by dynamic shifts in the epidemiological triad. Environmental factors, particularly those linked to ongoing climate change, have demonstrably altered the distribution of disease vectors [8] and potentially increased the incidence of diarrhea [9]. At the host level, there has been reduction in HIV- infection and mother-to child transmission, stunting, and wasting [10,11]. Additionally, targeted interventions such as the wider availability and use of Oral Rehydration Solution (ORS), upscaling of zinc [12] and the introduction of rotavirus vaccines [13] have reshaped the management and prevention of diarrheal diseases. Concurrently, the evolution of pathogens at the agent level has introduced new challenges, most notably the rise of antimicrobial resistance [14] and serotype distribution [15]. Moreover, several non-diarrhea-specific public health interventions and vaccination programs have been implemented. These include but are not limited to, RTS,S malaria vaccine, pneumococcal conjugate vaccine (PCV10), prevention of mother-to-child transmission initiatives, vitamin A supplementation, and distribution of insecticide-treated nets (ITNs). Understanding the patterns of diarrheal comorbidity and its associated factors against this backdrop of evolving epidemiological dynamics is imperative to inform the development of integrated strategies to effectively address this public health problem. Here, we leverage

data from three consecutive enteric studies to explore temporal patterns and determine risk factors of diarrheal comorbidity among children aged < 5 years in rural western Kenya.

## Materials and methods

### Ethics statement

The protocols for GEMS, VIDA, and EFGH studies were reviewed and approved by the Scientific and Ethical Review Unit (SERU) of the Kenya Medical Research Institute (KEMRI) with the approvals granted under KEMRI SSC Protocol #1155, SERU#2996 and SERU#4362, respectively. Additionally, GEMS and VIDA obtained approval from the Institutional Review Board (IRB) of the University of Maryland, School of Medicine with approvals granted under UMB Protocol #H-28327 for GEMS and UMB Protocol #: HM-HP-00062472 for VIDA. Moreover, the IRB for the Centers for Disease Control and Prevention, Atlanta, GA, USA approved the VIDA protocol (reliance agreement 6729), and formally deferred its review of the GEMS protocol to the University of Maryland IRB (CDC Protocol # 5038). Prior to initiating any study procedures, written informed consent was obtained from each caregiver of all participating children across all three studies.

### Study design

This is a secondary, pooled individual-participant data analysis with a retrospective cohort design, using data from three prospective enteric multicenter studies, (i) the Global Enteric Multicenter Study (GEMS, 2008–2012), (ii) the Vaccine Impact on Diarrhea in Africa (VIDA, 2015–2018), and (iii) the Enteric for Global Health (EFGH) *Shigella* surveillance study (2022–2024).

### Study setting and population

The three studies were conducted in a predominantly rural area of western Kenya, previously described by Odhiambo et al. [16] and more recently by Omore et al. [17], with a population of 217,952 as of 2023. The population has a high level of primary education attainment (88.8%) and primarily engages in subsistence farming as the main economic activity. Vaccination coverage rates in the study area surpassed national estimates (80.0%), with 88.1% of children aged 12–23 months having been fully vaccinated based on basic antigens: One dose of Bacillus Calmette–Guérin (BCG) vaccine, three doses of the poliomyelitis vaccine, three doses of the diphtheria-pertussis-tetanus (DPT) vaccine, and one dose of the measles-mumps-rubella (MMR) vaccine. Rotavirus vaccine was introduced into the national schedule in July 2014, with the first doses given at 6 and 10 weeks of age, using the Rotarix vaccine. The estimated national coverage for the last dose (second dose) of the rotavirus vaccine in 2024 was 82% [18]. However, Kenya has since switched its rotavirus vaccine from Rotarix (2 doses) to Rotavac (3 doses) starting in November 2022 due to global supply shortages of Rotarix [19]. Additionally, 69.9% of the children were fully vaccinated according to the national immunization schedule (basic antigens as well as a birth dose of OPV, a dose of IPV, three doses of the pneumococcal vaccine, and two doses of the rotavirus vaccine) [11]. Siaya County still faces challenges of child malnutrition with 19.2%, 1.7% and 7.0% of children aged < 5 years reported to be stunted, wasted and under-weight, respectively. The reported HIV prevalence of 14.3% in the county is approximately 3 times the national average [20]. In addition, the study area is situated within the lake endemic malaria zone, characterized by perennial intense malaria transmission and a prevalence rate of 27% (21).

### Data sources

The three primary studies (GEMS, VIDA and EFGH) have been previously described [22–26]. Briefly, GEMS was a 3-year prospective case-control study designed to assess the population-level burden, etiology, and clinical consequences of moderate-to-severe diarrhea (MSD) among children under five years of age in sub-Saharan Africa (SSA) and South Asia. Moderate-to-severe diarrhea (MSD) cases, defined as children aged 0–59 months presenting at a sentinel health

center with diarrhea (defined as ≥ 3 looser-than-normal stools within 24 hours) that began within the past 7 days after ≥7 diarrhea-free days and had ≥ 1 of the following: sunken eyes, poor skin turgor, dysentery, required intravenous rehydration, or hospitalization. Diarrhea-free controls matched by age, gender and geographical location were enrolled within 14 days of case enrolment. Following GEMS, the VIDA study employed the same design and methods to evaluate diarrheal etiologies, assess rotavirus vaccine effectiveness, and measure the population-level impact of rotavirus vaccine introduction among children <5 years residing in censused populations in 3 SSA countries. More recently, EFGH was conducted, using cross-sectional and longitudinal study designs, to establish incidence and consequences of *Shigella* medically attended diarrhea (MAD) among children aged 6–35 months within 7 country sites in Africa, Asia, and Latin America. Eligible MAD cases were children aged 6–35 months presenting at a sentinel health center with diarrhea (defined as ≥ 3 looser-than-normal stools within 24 hours) that began within the past 7 days after ≥2 diarrhea-free days. In this analysis, we restricted EFGH data to children who met the GEMS/VIDA MSD criteria (diarrhea with dehydration, dysentery, or requiring hospitalization). The authors did not have access to any information that could identify individual participants during or after data collection.

In all the three studies, trained study personnel collected comprehensive data at enrollment, encompassing demographic information, illness history, anthropometric measurements, clinical features and the socio-economic characteristics of the enrolled children. Study clinicians made diagnoses based on clinical history and physical exam.

## Outcome variable

Comorbidity count was treated as a count outcome variable based on Integrated Management of Childhood Illnesses (IMCI) case definitions [27] and clinician diagnoses at enrolment. We assessed ten comorbidities that included: malaria, other invasive bacterial infection, pneumonia, severe acute malnutrition (SAM), meningitis, acute febrile illness (AFI), respiratory Illness (non-pneumonia), anemia, stunting, and wasting. The detailed definitions of each comorbidity are shown in S1 Table.

## Independent variables

A comprehensive list of demographic, clinical features, water and sanitation infrastructure, and the socio-economic characteristics were assessed as potential covariates. We used the *GEMS-modified Vesikari score system* [28], a 17-point scoring tool adapted from the original 20-point scale developed by Ruuska and Vesikari. The main difference is that vomiting is captured as a simple yes/no variable, rather than being assessed by frequency and duration (days). We also included full vaccination status, PCV10 vaccination status, and rotavirus vaccination status as covariates. These data were obtained from the Health Demographic Surveillance System for GEMS and VIDA while EFGH directly collected vaccination data. Full vaccination was defined as receipt of all eight doses recommended under the Expanded Programme on Immunization (EPI): one dose of Bacille Calmette–Guérin (BCG), three doses each of pentavalent and oral polio vaccines, and one dose of measles vaccine. PCV10 vaccination status was coded as unvaccinated or vaccinated (≥1 dose) based on immunization card records, with a separate category for unknown when immunization cards were unavailable. Similarly, rotavirus vaccination status was coded as pre-vaccine for the GEMS study period, during which the vaccine was not available, and as unvaccinated or vaccinated (≥1 dose) based on immunization card records during the VIDA and EFGH studies, with an unknown category when immunization cards were unavailable. These categories were modeled explicitly to distinguish structural absence of vaccination data from missing verification within the vaccine era. Additionally, a continuous variable representing time (year) and climate covariates (monthly rainfall and temperature) were also included. Monthly daytime land surface temperature (LSTD) and nighttime land surface temperature (LSTN) data were extracted from the Moderate Resolution Imaging Spectroradiometer (MODIS) [29]. Monthly rainfall data were processed from the Climate Hazards Group InfraRED Precipitation with Station data (CHIRPS) [30] while historical daily near-surface air temperature data were extracted from the ERA5-Land and processed to a monthly scale [31]. The above monthly climate data

were extracted using defined bounding boxes representing the study catchment area. We also included climate variables lagged by up to three months.

## Statistical analysis

We summarized categorical variables by generating frequency tables with corresponding percentages while continuous variables were summarized using the median and interquartile range (IQR) for each study and for the combined dataset.

We calculated the annual frequency and prevalence of children with 0, 1, 2, 3, and ≥4 comorbid conditions. Prevalence was computed by dividing the number of children in each comorbidity group by the total number of enrolled children per year. To evaluate temporal trends in the prevalence of individual comorbidities and any comorbidity, we conducted a Cochran-Armitage trend test. To assess how comorbidity burden varied by age and study period, we categorized children into age groups (0–11 months, 12–23 months, 24–59 months) and calculated the prevalence of each comorbidity category within these strata as well as overall. We explored temporal trends for 10 individual comorbidities and any comorbidity with comorbidity prevalence plotted over time with separate trajectories shown for each condition and study period. To evaluate patterns of co-occurrence among comorbidities, we generated an UpSet plot showing intersecting comorbidity profiles.

We fitted generalized estimating equations (GEE) with a Poisson distribution and a log link, to determine the risk factors of comorbidity count [32]. Initially, bivariate models were fitted to assess the unadjusted association between each predictor variable. All variables were included in the multivariable model except the Vesikari score, as it is a composite of other predictors. The primary analysis focused on children aged 0–59 months and included all covariates (including full vaccination status and PCV10 vaccination status) except rotavirus vaccination. To assess whether differences in age distributions between EFGH and GEMS/VIDA influenced the results, we conducted a sensitivity analysis restricting the sample to children aged 6–35 months and fitting a model otherwise identical to the primary model. We conducted an additional sensitivity analysis to evaluate the impact of rotavirus vaccination by including rotavirus vaccination status and calendar year as a continuous variable. Because rotavirus vaccination was introduced during the later study years, calendar year and vaccination status were perfectly confounded when year was modeled categorically; therefore, modeling year as a continuous variable allowed us to account for secular trends while enabling estimation of the vaccination effect. Potential collinearity among the selected predictors was assessed using the Cramer's V statistic. To account for within-subject correlation and potential heteroscedasticity, we used an exchangeable correlation structure and specified robust standard errors via the Huber-White sandwich estimator. We reported the incidence rate ratios (IRRs) and their 95% confidence intervals (CIs). To assess the adequacy of the Poisson model, we calculated the overdispersion statistic as the ratio of the Pearson chi-square statistic to the residual degrees of freedom. A ratio substantially greater than 1 was considered evidence of overdispersion. In our analysis, no evidence of overdispersion was observed. All the statistical analyses were carried out using R software, version 4.4.1 (R Foundation for Statistical Computing, Vienna, Austria).

## Results

### Baseline characteristics

A total of 4,148 children with diarrhea were analyzed, drawn from the three studies: GEMS (n = 1,778), VIDA (n = 1,554), and EFGH (n = 816). The cohort's median age was 14 months (IQR: 8–24), and 44.6% (n = 1,850) were female. The baseline characteristics of patients overall and stratified by study are shown in Table 1. Caregiver education below the primary level was reported for 33.8% (n = 1,400) of participants overall, with a downward trend from 44.8% in GEMS to 9.7% in EFGH. Access to improved water and sanitation was reported for 65.6% (n = 2,720) and 38.3% of households, respectively, with improved sanitation showing an increasing trend across studies. Clinically, the median diarrhea duration was five days (IQR: 3–8). Over half of the children had seven or more episodes of diarrhea (56.8%) and vomiting (52.3%,

**Table 1. Characteristics of children aged<59 months presenting with diarrhea in Western Kenya, 2008-2024.**

| Variable | Category | Overall (N=4,732) | GEMS (N=1,778) | VIDA (N=1,554) | EFGH (N=816) |
|---|---|---|---|---|---|
| | | n (%) | n (%) | n (%) | n (%) |
| **Socio-demographic** | | | | | |
| Age | Median [IQR] | 14 [8-24] | 12 [7-24] | 15 [9-25] | 13 [9-20] |
| Age Categories | 0_11 m | 1753 (42.3) | 829 (46.6) | 586 (37.7) | 338 (41.4) |
| | 12_23 m | 1352 (32.6) | 491 (27.6) | 528 (34.0) | 333 (40.8) |
| | 24_59 m | 1043 (25.1) | 458 (25.8) | 440 (28.3) | 145 (17.8) |
| Gender | Female | 1850 (44.6) | 768 (43.2) | 707 (45.5) | 375 (46.0) |
| Caregiver education less than primary school | Yes | 1400 (33.8) | 796 (44.8) | 525 (33.8) | 79 (9.7) |
| **Water and Sanitation** | | | | | |
| Improved water | Unimproved | 1428 (34.4) | 691 (38.9) | 447 (28.8) | 290 (35.5) |
| | Improved | 2720 (65.6) | 1087 (61.1) | 1107 (71.2) | 526 (64.5) |
| Improved sanitation | Unimproved | 2559 (61.7) | 1374 (77.3) | 851 (54.8) | 334 (40.9) |
| | Improved | 1589 (38.3) | 404 (22.7) | 703 (45.2) | 482 (59.1) |
| Respiratory Rate | Median [IQR] | 36 [31-42] | 37.5 [31-45] | 36.5 [31.5-41] | 34 [29-39] |
| Respiratory Rate categories | Normal | 3498 (84.3) | 1433 (80.6) | 1352 (87.0) | 713 (87.4) |
| | Low | 151 (3.6) | 59 (3.3) | 38 (2.4) | 54 (6.6) |
| | High | 499 (12.0) | 286 (16.1) | 164 (10.6) | 49 (6.0) |
| Temperature | Median [IQR] | 37.1 [36.5-38] | 37.2 [36.6-38.2] | 36.8 [36.4-37.6] | 37.2 [36.7-38.1] |
| Temperature categories | <37.1 | 2084 (50.8) | 761 (42.8) | 948 (61.0) | 375 (48.5) |
| | 37.1-38.4 | 1349 (32.9) | 639 (36.0) | 435 (28.0) | 275 (35.6) |
| | 38.5-38.9 | 267 (6.5) | 144 (8.1) | 87 (5.6) | 36 (4.7) |
| | ≥39.0 | 404 (9.8) | 233 (13.1) | 84 (5.4) | 87 (11.3) |
| Max no. of watery diarrhea episodes in a day, n (%) | ≤ 6 | 1793 (43.2) | 1319 (74.2) | 283 (18.2) | 191 (23.4) |
| | ≥7 | 2355 (56.8) | 459 (25.8) | 1271 (81.8) | 625 (76.6) |
| Pre-enrolment diarrhea days | Median [IQR] | 3 [2-3] | 3 [2-3] | 3 [2-4] | 2 [2-3] |
| Experienced vomiting | Yes | 2170 (52.3) | 858 (48.3) | 884 (56.9) | 428 (52.5) |
| Dehydration | None | 224 (5.4) | 64 (3.6) | 84 (5.4) | 76 (9.3) |
| | Severe | 902 (21.7) | 452 (25.4) | 415 (26.7) | 35 (4.3) |
| | Some | 3022 (72.9) | 1262 (71.0) | 1055 (67.9) | 705 (86.4) |
| Admitted | Yes | 401 (9.7) | 193 (10.9) | 155 (10.0) | 53 (6.5) |
| Diarrhea Duration (Days) | Median [IQR] | 5 [3-8] | 7 [5-9] | 5 [3-7] | 3 [2-5] |
| **Prolonged diarrhea**£ | Yes | 1288 (32.6) | 743 (47.1) | 421 (27.1) | 124 (15.2) |
| Persistent diarrhea€ | Yes | 162 (4.1) | 94 (6.0) | 58 (3.7) | 10 (1.2) |
| Vesikari score¥ | Median [IQR] | 10 [8-11] | 9 [7-11] | 11 [9-12] | 9 [7-11] |
| Vesikari score Categories | Mild | 360 (8.7) | 221 (12.4) | 54 (3.5) | 85 (10.4) |
| | Moderate | 2261 (54.5) | 1077 (60.6) | 691 (44.5) | 493 (60.4) |
| | Severe | 1527 (36.8) | 480 (27.0) | 809 (52.1) | 238 (29.2) |
| Fully vaccinated§ | No | 1379 (33.2) | 580 (32.6) | 550 (35.4) | 249 (30.5) |
| | Yes | 1804 (43.5) | 530 (29.8) | 839 (54.0) | 435 (53.3) |
| | Unknownᵗ | 965 (23.3) | 668 (37.6) | 165 (10.6) | 132 (16.2) |
| PCV10 vaccine | Unvaccinated | 59 (1.4) | 17 (1.0) | 33 (2.1) | 9 (1.1) |
| | Vaccinated (≥ 1) | 2268 (54.7) | 238 (13.4) | 1355 (87.2) | 675 (82.7) |
| | Unknownᵗ | 1821 (43.9) | 1523 (85.7) | 166 (10.7) | 132 (16.2) |

*(Continued)*

**Table 1.** (Continued)

| Variable | Category | Overall (N=4,732) | GEMS (N=1,778) | VIDA (N=1,554) | EFGH (N=816) |
|---|---|---|---|---|---|
| | | n (%) | n (%) | n (%) | n (%) |
| Rotavirus Vaccine | Pre-vaccine | 1778 (42.9) | 1778 (100.0) | – | – |
| | Vaccinated (≥ 1) | 1756 (42.3) | – | 1213 (78.1) | 543 (66.5) |
| | Unvaccinated | 368 (8.9) | – | 227 (14.6) | 141 (17.3) |
| | Unknownᵗ | 246 (5.9) | – | 114 (7.3) | 132 (16.2) |
| **Climate Variables** | | | | | |
| Rainfall (mm) | Median [IQR] | 120.9 [75.1-178.8] | 114.8 [70.3-159.6] | 102.5 [52.2-179.7] | 139.4 [88-181.9] |
| Rain (mm) tercile | Tercile 1:13.92-<82.49 | 1383 (33.3) | 610 (34.3) | 573 (36.9) | 200 (24.5) |
| | Tercile 2:82.49-<158.30 | 1383 (33.3) | 633 (35.6) | 512 (32.9) | 238 (29.2) |
| | Tercile 3:158.30-<353.40 | 1382 (33.3) | 535 (30.1) | 469 (30.2) | 378 (46.3) |
| LSTD (°C) | Median [IQR] | 31.9 [29.8-34.8] | 31.7 [30-35.5] | 33.2 [30.1-35.3] | 30.3 [28.7-32.8] |
| LSTD (°C) tercile | Tercile 1:26.36-<30.09 | 1383 (33.3) | 544 (30.6) | 411 (26.4) | 428 (52.5) |
| | Tercile 2:30.09-<32.96 | 1383 (33.3) | 674 (37.9) | 405 (26.1) | 304 (37.3) |
| | Tercile 3:32.96-<41.38 | 1382 (33.3) | 560 (31.5) | 738 (47.5) | 84 (10.3) |
| **Comorbidities** | | | | | |
| At least 1 comorbidity | No | 402 (9.7) | 126 (7.1) | 167 (10.7) | 109 (13.4) |
| | Yes | 3746 (90.3) | 1652 (92.9) | 1387 (89.3) | 707 (86.6) |
| Number of comorbidities | Median [IQR] | 2 [1-2] | 2 [1-2] | 1 [1-2] | 1 [1-2] |
| Malaria | Yes | 2000 (48.3) | 1047 (58.9) | 613 (39.6) | 340 (41.7) |
| Pneumonia | Yes | 2053 (49.5) | 1020 (57.4) | 823 (53.0) | 210 (25.7) |
| Stunted | Yes | 1018 (24.7) | 491 (27.8) | 372 (24.1) | 155 (19.1) |
| Wasted | Yes | 443 (10.7) | 354 (20.0) | 58 (3.7) | 31 (3.8) |
| AFI | Yes | 404 (9.7) | 104 (5.8) | 194 (12.5) | 106 (13) |
| Respiratory illness | Yes | 207 (5) | 4 (0.2) | 43 (2.8) | 160 (19.6) |
| Bacterial Infection | Yes | 289 (7) | 81 (4.6) | 143 (9.2) | 65 (8) |
| Severe Acute Malnutrition (SAM) | Yes | 186 (4.5) | 121 (6.8) | 54 (3.5) | 11 (1.3) |
| Anemia | Yes | 115 (2.8) | 26 (1.5) | 47 (3.0) | 42 (5.1) |
| Meningitis | Yes | 1 (0) | 1 (0.1) | 0 (0) | 0 (0) |

AFI-Acute Febrile illness; LSTD- Land Surface Temperature – Daytime.

β Cutoffs for respiratory rate based on the Pediatric Advanced Life Support (PALS) guidelines.

£ ≥ 7 days of diarrhea during index diarrhea episode.

€ ≥ 14 days of diarrhea during index diarrhea episode.

¥ Modified 17-point Vesikari score.

§ Full vaccinated- 1 dose of BCG, 3 doses of pentavalent and polio, and 1 dose of measles.

ᵗ Unknown vaccination card not available.

n=2,170), while 9.7% (n=401) required admission. Based on a 17-point modified vesikari score, disease severity was moderate overall (median: 10; IQR: 8–11), with VIDA study having the most severe episodes (median: 11; IQR: 9–12) compared to GEMS and EFGH (median: 9; IQR: 7–11). The proportion of fully vaccinated children increased from GEMS (530; 29.8%) to VIDA (839; 54.0%) and remained stable in EFGH (435; 53.3%). The proportion of children vaccinated with PCV10 also had an increase from GEMS (238; 13.4%) to VIDA (1355; 87.2%) but showed a slight drop in EFGH (675; 82.7%). Rotavirus vaccination, which was introduced in 2014, showed the highest coverage in VIDA (1,213; 78.1%) and a lower prevalence in EFGH (543; 66.5%).

## Patterns of comorbidities

Majority of the children (90.3%, n = 3,746) had at least one comorbidity, with a declining trend across studies: GEMS (92.9%, n = 1,652), VIDA (89.3%, n = 1,387), and EFGH (86.6%, n = 707). The most common comorbidities were pneumonia (49.5%, n = 2,053), malaria (48.3%, n = 2,000), and stunting (24.7%, n = 1,018), with pneumonia and stunting declining over time across the three study periods (Table 1). Moreover, we observed distinct age-related patterns in comorbidity prevalence across the three study cohorts. In GEMS, which had the highest comorbidity burden, disease complexity peaked in younger children aged 12–23 months, who exhibited the highest prevalence of multiple comorbidities (≥2 [66.0%]). Conversely, the VIDA cohort had an intermediate burden and the EFGH cohort having the least comorbidity burden, and demonstrated a more linear trend. In VIDA the prevalence of having two or more comorbidities rose from 44% in the 0–11 month age group to 49% in the 24–59 month group. A similar, though less pronounced, increase was seen in the EFGH cohort, rising from 37% to 43% across the same age brackets (Fig 1).

Over the 16-year period from 2008 to 2024, the burden of comorbidities has shown a shift towards fewer comorbidities per child, particularly a rise in children with exactly one comorbidity. The prevalence of patients having a single comorbidity increased, from 28.9% in 2008 to approximately half (49.7%) in 2024. Concurrently, there were declines in the prevalence of multiple comorbidities between 2008 and 2024, with reductions observed among children with two (38.8% to 30.1%), three (21.8% to 4.6%), and four or more comorbidities (7.7% to 2.0%). While the proportion with zero comorbidities showed some year-to-year fluctuation, but it generally increased compared to the earlier years (2.7% in 2008 vs. 13.7% in 2024) (Fig 2).

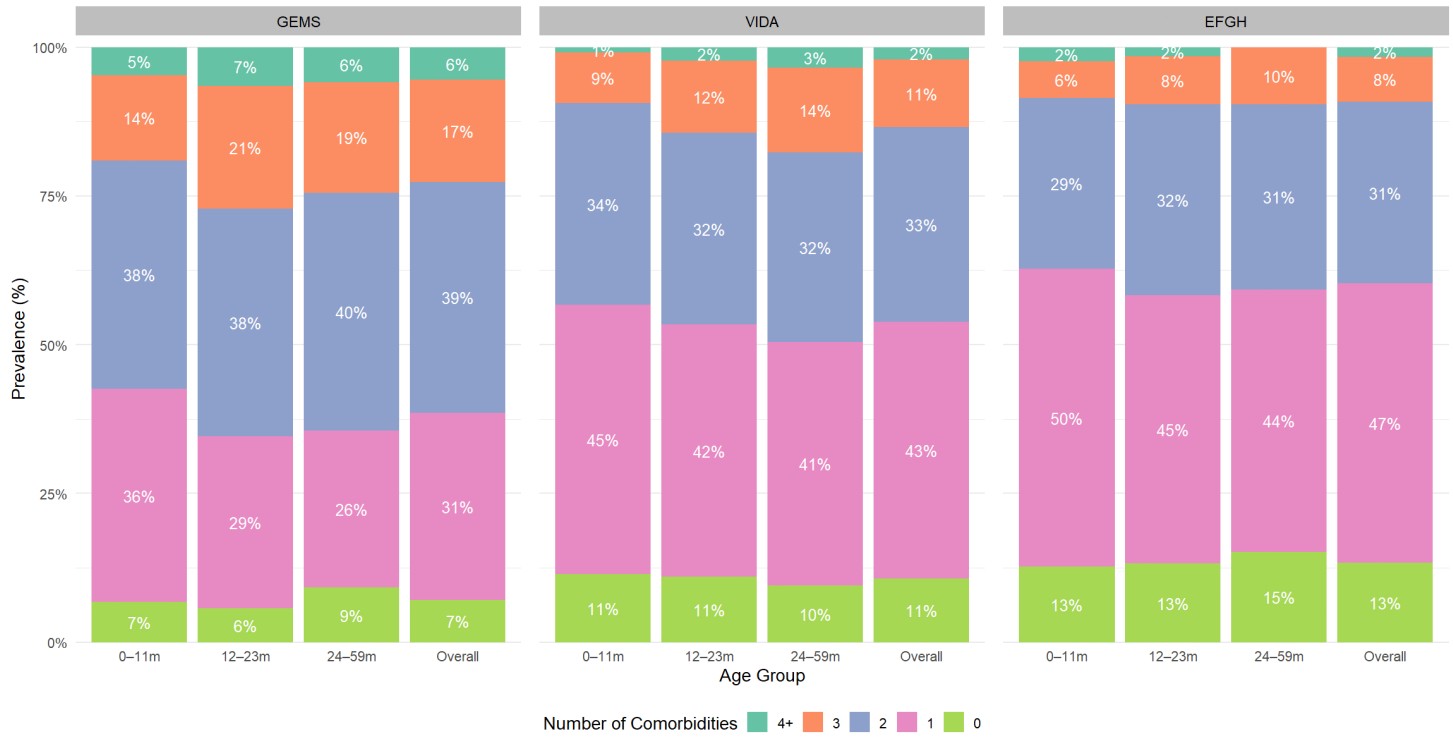

**Fig 1. Age-stratified prevalence of number of comorbidities in childhood diarrhea in Western Kenya, 2008-2024.**

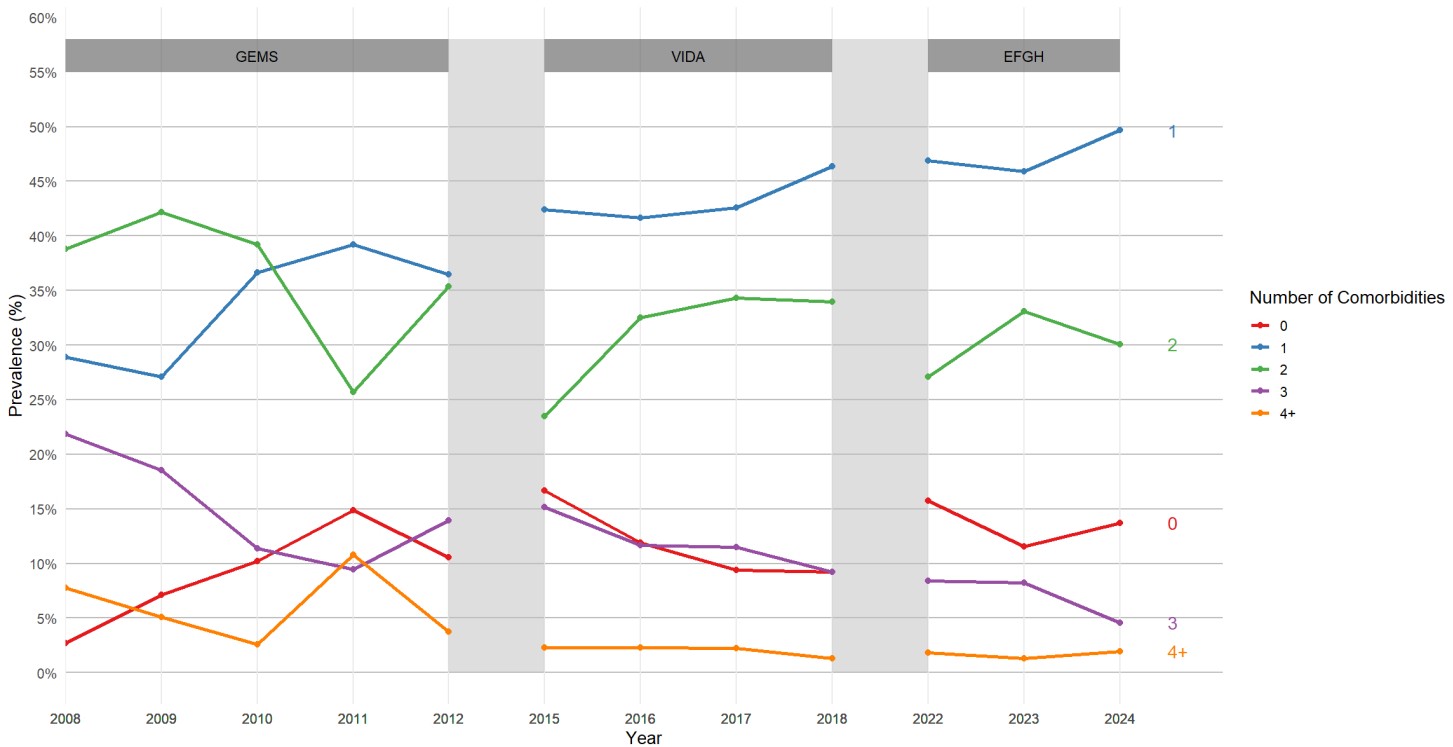

**Fig 2. Prevalence of number of comorbidities in childhood diarrhea in Western Kenya, 2008-2024.**

During the study period, we observed a distinct epidemiological transition. Traditional major comorbidities like malaria, pneumonia, stunting, wasting, and SAM have shown significant reductions over the 16-year period (Fig 3 and Table 2).

Specifically, malaria decreased by nearly half from 65.4% in 2008 to 35.9% in 2024, while pneumonia declined from 65.2% to 28.8% over the same period. Similarly, the prevalence of stunting dropped from 30.5% to 23.0%, and wasting showed an even steeper reduction from 22.3% to just 2.6%. Cases of SAM also declined steadily from 8.7% in 2008 to 2.0% in 2024. On the other hand, respiratory illness (non-pneumonia), which was nearly absent in 2008 (0.2%), rose to peaks in 2022–2023 (>20%) before slightly decreasing to 14.4% in 2022 emerging as a new concern or a less severe manifestation of the causes of pneumonia which has decreased. Acute febrile illness (AFI) increased from 4.4% to peaks in 2017 (14.8%) and 2023 (15.1%), before slightly decreasing to 11.8% in 2024. Additionally, anemia also rose slightly from 1.0% in 2008 to 3.4% in 2024, with a peak of 6.2% in 2022. Bacterial infections showed fluctuating trends, ranging from 1.4% in 2010 to around 7–10% in more recent years. All observed trends were statistically significant (p < 0.01), indicating meaningful changes in the comorbidity profiles of children with diarrhea over time (Table 2).

### Co-occurrence

Single comorbidities were the most common patient groupings with pneumonia alone reported in 566 (15.1%), malaria alone in 561 (15.0%), AFI alone in 218 (5.8%), and stunting alone in 107 (2.9%) —these being the four most frequent single comorbidities (Fig 4). We observed 32 distinct comorbidity co-occurrences with most common combinations involving malaria, pneumonia and stunting. Specifically, malaria and pneumonia was the most common

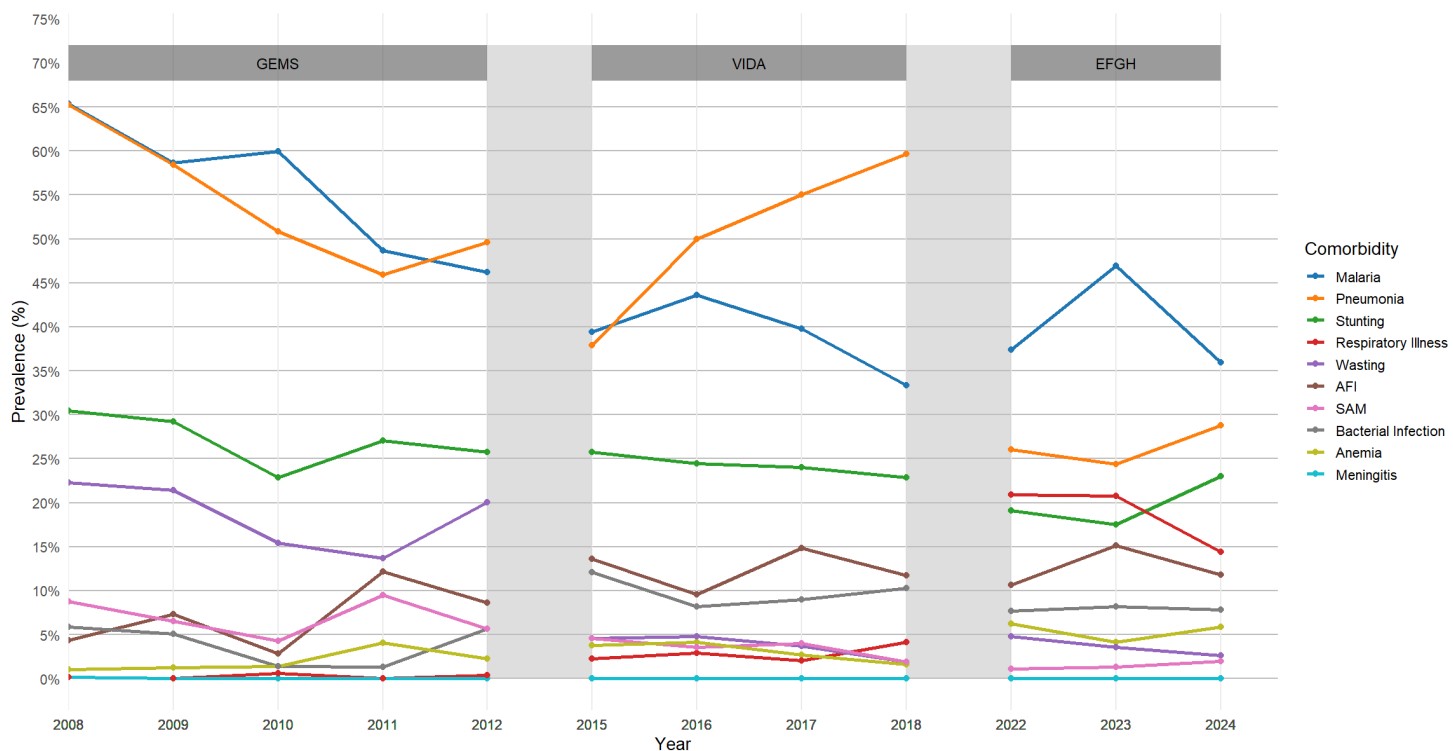

**Fig 3. Prevalence of individual comorbidities in childhood diarrhea in Western Kenya, 2008-2024.**

combination 644 (17.2%), followed by pneumonia and stunting 161 (4.3%), the triad of malaria, pneumonia and stunting 158 (4.2%), and malaria and stunting 138 (3.7%). Although stunting alone was not the most prevalent single comorbidity, it was a frequent component in the most common disease combinations, highlighting its central role in overall patient morbidity.

### Risk factors for diarrheal comorbidity

Based on our multivariable model, we observed that older children had a significantly higher number of comorbidities compared to infants (24–59 months: adjusted Incidence Rate Ratio (aIRR)= 1.08, 95% CI [1.02–1.13]) (Table 3). Furthermore, children of caregivers having less than a primary school education had a 9% higher number of comorbidities compared to those with caregivers having secondary or higher education (aIRR = 1.09, 95%CI [1.05–1.14]). Similarly, dehydration was associated with 25% and 35% higher number of comorbidities for some dehydration (aIRR = 1.25, 95% CI [1.12-1.40]) and severe dehydration (aIRR = 1.35, 95%CI [1.20–1.52]), respectively. Children with unknown vaccination status due to unavailable immunization cards had an 8% higher number of comorbidities compared with children who were not fully vaccianted (aIRR = 1.08, 95% CI: 1.01–1.15). Other clinical signs associated with a significantly higher number of comorbidities included vomiting (aIRR: 1.25, 95% CI [1.19–1.31]) and a high respiratory rate (aIRR = 1.21, 95%CI [1.14–1.28]) (Table 3). While three-month lagged rainfall and land surface temperatures were associated with 7% and 8% higher number of comorbidities in Terciles 3 (Rainfall Tercile 3: aIRR: 1.07, 95% CI [1.00–1.15]; LSTD Tercile 3: aIRR: 1.08, 95% CI [1.00–1.17]), respectively.

Conversely, female gender was associated with 5% fewer comorbidities compared to males (aIRR = 0.95, 95%CI [0.91–0.99]). Importantly, there was a significant and consistent downward temporal trend across the study period.

**Table 2. Temporal trends of diarrheal comorbidities among children aged <59 months in Western Kenya, 2008-2024.**

| Comorbidity | 2008 | 2009 | 2010 | 2011 | 2012 | 2015 | 2016 | 2017 | 2018 | 2022 | 2023 | 2024 | p_value for Trend |
|---|---|---|---|---|---|---|---|---|---|---|---|---|---|
| Any Comorbidity | 579 (97.3%) | 456 (92.9%) | 316 (89.8%) | 63 (85.1%) | 238 (89.5%) | 110 (83.3%) | 423 (88.1%) | 568 (90.6%) | 286 (90.8%) | 230 (84.2%) | 345 (88.5%) | 132 (86.3%) | <0.001 |
| Malaria | 389 (65.4%) | 288 (58.7%) | 211 (59.9%) | 36 (48.6%) | 123 (46.2%) | 52 (39.4%) | 208 (43.6%) | 249 (39.8%) | 104 (33.3%) | 102 (37.4%) | 183 (46.9%) | 55 (35.9%) | <0.001 |
| Pneumonia | 388 (65.2%) | 287 (58.5%) | 179 (50.9%) | 34 (45.9%) | 132 (49.6%) | 50 (37.9%) | 240 (50.0%) | 345 (55.0%) | 188 (59.7%) | 71 (26%) | 95 (24.4%) | 44 (28.8%) | <0.001 |
| Stunting | 180 (30.5%) | 143 (29.2%) | 80 (22.9%) | 20 (27%) | 68 (25.8%) | 34 (25.8%) | 117 (24.5%) | 149 (24.0%) | 72 (22.9%) | 52 (19.1%) | 68 (17.5%) | 35 (23%) | <0.001 |
| Wasting | 132 (22.3%) | 105 (21.4%) | 54 (15.4%) | 10 (13.7%) | 53 (20.1%) | 6 (4.5%) | 23 (4.8%) | 23 (3.7%) | 6 (1.9%) | 13 (4.8%) | 14 (3.6%) | 4 (2.6%) | <0.001 |
| AFI | 26 (4.4%) | 36 (7.3%) | 10 (2.8%) | 9 (12.2%) | 23 (8.6%) | 18 (13.6%) | 46 (9.6%) | 93 (14.8%) | 37 (11.7%) | 29 (10.6%) | 59 (15.1%) | 18 (11.8%) | <0.001 |
| Respiratory Illness | 1 (0.2%) | 0 (0%) | 2 (0.6%) | 0 (0%) | 1 (0.4%) | 3 (2.3%) | 14 (2.9%) | 13 (2.1%) | 13 (4.1%) | 57 (20.9%) | 81 (20.8%) | 22 (14.4%) | <0.001 |
| Bacterial Infection | 35 (5.9%) | 25 (5.1%) | 5 (1.4%) | 1 (1.4%) | 15 (5.6%) | 16 (12.1%) | 39 (8.2%) | 56 (8.9%) | 32 (10.3%) | 21 (7.7%) | 32 (8.2%) | 12 (7.8%) | <0.001 |
| SAM | 52 (8.7%) | 32 (6.5%) | 15 (4.3%) | 7 (9.5%) | 15 (5.6%) | 6 (4.5%) | 17 (3.5%) | 25 (4.0%) | 6 (1.9%) | 3 (1.1%) | 5 (1.3%) | 3 (2.0%) | <0.001 |
| Anemia | 6 (1%) | 6 (1.2%) | 5 (1.4%) | 3 (4.1%) | 6 (2.3%) | 5 (3.8%) | 20 (4.2%) | 17 (2.7%) | 5 (1.6%) | 17 (6.2%) | 16 (4.1%) | 9 (5.9%) | <0.001 |
| Meningitis | 1 (0.2%) | 0 (0%) | 0 (0%) | 0 (0%) | 0 (0%) | 0 (0%) | 0 (0%) | 0 (0%) | 0 (0%) | 0 (0%) | 0 (0%) | 0 (0%) | -- |

AFI- Acute Febrile illness; SAM- Severe Acute Malnutrition.

Compared to the reference year 2008, the rate of comorbidities decreased progressively over time. However, estimates for 2009, 2011, and 2024 were not statistically significant. In the most recent years (2022–2023), rates were approximately 20–23% lower (2022: aIRR = 0.77, 95% CI [0.70–0.84]; 2023: aIRR = 0.80, 95% CI [0.71–0.90], 2024) (Table 3).

We observed largely consistent results when the analysis was restricted to children aged 6–35 months across all studies. The main additional findings were that PCV10 vaccination was associated with 16% fewer number of comorbidities compared with unvaccinated children (aIRR = 0.84, 95% CI: 0.73–0.97). In addition, three-month lagged land surface temperatures were associated with higher comorbidity counts, with estimates of 7% and 10% observed in terciles 2 and 3, respectively (Tercile 2: aIRR = 1.07, 95% CI: 1.00–1.14; Tercile 3: aIRR = 1.10, 95% CI: 1.01–1.20) (Table 4).

Similarly, inclusion of rotavirus vaccination in the model did not alter the overall findings. The key additional results were that access to improved water was associated with a 4% fewer number of comorbidities compared with unimproved water sources (aIRR = 0.96, 95% CI: 0.92–0.99). Moreover, modeling year as a continuous variable showed a steady temporal decline in comorbidity burden, with an estimated 3% reduction in comorbidity count per year relative to 2008 (aIRR = 0.97, 95% CI: 0.96–0.99; S2 Table). More specifically, relative to the pre–rotavirus vaccination era (GEMS, 2008–2012), children who were vaccinated, unvaccinated, or had unknown vaccination status for rotavirus due to unavailable immunization cards showed a 10–20% lower comorbidity count in univariable analyses (vaccinated: IRR = 0.80, 95% CI: 0.77–0.83; unvaccinated: IRR = 0.80, 95% CI: 0.75–0.86; unknown: IRR = 0.90, 95% CI: 0.83–0.97) (S2 Table). However, these associations were not sustained after adjustment in the multivariable model.

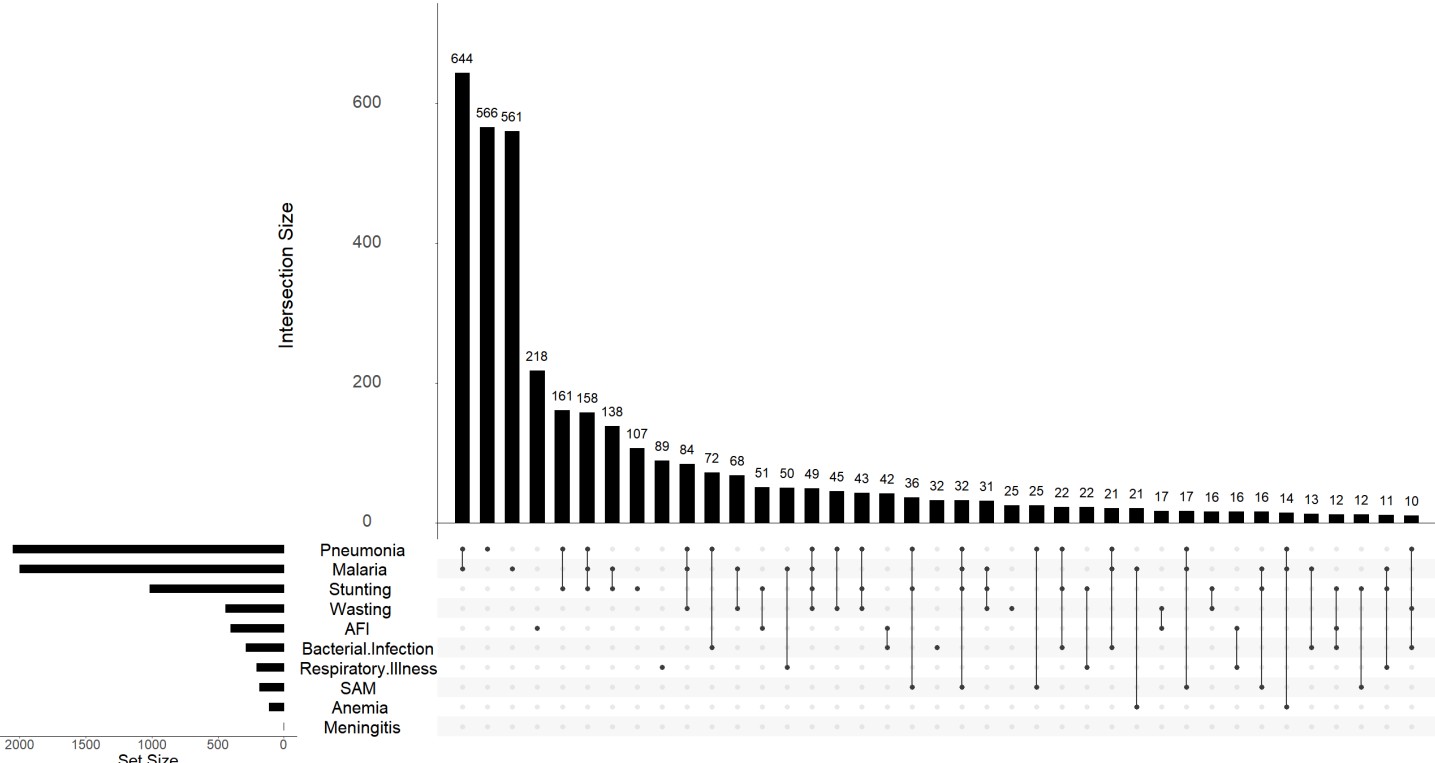

**Fig 4. Upset plot of comorbidity co-occurrence in childhood diarrhea in Western Kenya, 2008-2024.**

## Discussion

Over the 16-year period (2008–2024), the burden and profile of comorbidities observed among children with diarrhea in Kenya has shifted significantly. While comorbidities remain common (90.3% overall), there is a clear decline in both prevalence and severity over time and across successive studies. Pneumonia, malaria, and stunting were the most prevalent single comorbidities as well as the most common co-occurrence combinations. Additionally, there has been a shift toward fewer comorbidities per child, with an increase in children presenting with only one comorbidity and a decline in those with multiple conditions (≥ 2 comorbidities). Moreover, traditional comorbidities (malaria, pneumonia, wasting, and SAM) have declined significantly, while respiratory illness (non-pneumonia), AFI, and anemia have emerged or increased. Stunting frequently co-occurred with other conditions, underlining its role in compounded morbidity. Finally, multivariable analysis identified older age, lower caregiver education, dehydration, vomiting, high respiratory rate, and three month lagged rainfall and land surface temperature as significant predictors of higher comorbidity counts, while female sex was associated with fewer comorbidities. Importantly, comorbidity burden decreased significantly over time, with 20–23% fewer comorbidities in recent years (2022–2023), reflecting a broader epidemiological transition.

The observed decline in the prevalence and severity of diarrheal comorbidities over time may reflect broader improvements in child health, nutrition, and healthcare systems. The likely most important driver of this observation is the success of targeted public health interventions and vaccination programs that directly target the most common comorbidities observed. Furthermore, reduced mother-to-child transmission of HIV in this setting may also explain in part the reduction of the burden of diarrheal comorbidity observed. Moreover, the scale-up of prevention of mother-to-child transmission (PMTCT) programs in Kenya (over 90% coverage) has likely contributed to the decline in diarrheal comorbidity by

**Table 3. Factors associated with number of comorbidities in children aged 0-59 months with moderate-to-severe diarrhea in Western Kenya, 2008-2024.**

| Variable | Category | Bivariate | | Multivariable |
|---|---|---|---|---|
| | | Unadjusted IRR [95% CI] | P-value | Adjusted IRR [95% CI] |
| Age Categories | 0_11 m | Ref | | Ref |
| | 12_23 m | **1.05 [1.01-1.09]** | **0.011** | 1.03 [0.98-1.08] |
| | 24_59 m | **1.07 [1.02-1.12]** | **0.002** | **1.08 [1.02-1.13]** |
| Gender | Female | **0.95 [0.91-0.98]** | **0.003** | **0.95 [0.91-0.99]** |
| Caregiver education less than primary school | Yes | **1.14 [1.1-1.19]** | **<0.001** | **1.09 [1.05-1.14]** |
| Improved water | Unimproved | Ref | | Ref |
| | Improved | **0.94 [0.91-0.97]** | **<0.001** | 0.96 [0.92-1.00] |
| Improved sanitation | Unimproved | Ref | | Ref |
| | Improved | **0.94 [0.91-0.97]** | **0.001** | 1.01 [0.96-1.05] |
| Respiratory Rate categories[ß] | Normal | Ref | | Ref |
| | Low | **0.86 [0.78-0.94]** | **0.002** | 0.92 [0.81-1.04] |
| | High | **1.27 [1.22-1.33]** | **<0.001** | **1.21 [1.14-1.28]** |
| Max no. of watery diarrhea episodes in a day, n (%) | ≤ 6 | Ref | | Ref |
| | ≥7 | **0.91 [0.88-0.94]** | **<0.001** | 1.01 [0.96-1.05] |
| Pre-enrolment diarrhea days | | 1.01 [0.99-1.02] | 0.211 | – |
| Experienced vomiting | | **1.24 [1.19-1.29]** | **<0.001** | **1.25 [1.19-1.31]** |
| Dehydration | None | Ref | | Ref |
| | Severe | **1.46 [1.32-1.62]** | **<0.001** | **1.35 [1.20-1.52]** |
| | Some | **1.28 [1.15-1.42]** | **<0.001** | **1.25 [1.12-1.4]** |
| Vesikari score Categories[¥] | Mild | Ref | | Ref |
| | Moderate | **1.27 [1.17-1.38]** | **<0.001** | – |
| | Severe | **1.53 [1.41-1.66]** | **<0.001** | – |
| Fully vaccinated[§] | No | Ref | | Ref |
| | Yes | 0.96 [0.92-1.00] | 0.062 | 1.01 [0.96-1.06] |
| | Unknown[ŧ] | **1.11 [1.06-1.16]** | **<0.001** | **1.08 [1.01-1.15]** |
| PCV10 vaccine | Unvaccinated | Ref | | Ref |
| | Vaccinated (≥ 1) | 0.9 [0.78-1.03] | 0.116 | 0.90 [0.77-1.04] |
| | Unknown[ŧ] | 1.09 [0.95-1.25] | 0.232 | 0.93 [0.79-1.09] |
| Rainfall (mm) tercile | Tercile 1:13.92-<82.49 | Ref | | Ref |
| | Tercile 2:82.49-<158.30 | 1.01 [0.96-1.05] | 0.814 | 1.02 [0.94-1.09] |
| | Tercile 3:158.30-<353.40 | 0.99 [0.94-1.03] | 0.537 | 1.01 [0.94-1.10] |
| Rainfall 3-month lag (mm) tercile | Tercile 1:13.92-<82.49 | Ref | | Ref |
| | Tercile 2:82.49-<158.30 | 1.05 [0.99-1.11] | 0.052 | 1.06 [0.99-1.13] |
| | Tercile 3:158.30-<353.40 | 1.04 [0.98-1.09] | 0.188 | **1.07 [1.00-1.15]** |
| LSTD (°C) tercile | Tercile 1:26.36-<30.09 | Ref | | Ref |
| | Tercile 2:30.09-<32.96 | 0.98 [0.94-1.02] | 0.390 | 1.00 [0.94-1.05] |
| | Tercile 3:32.96-<41.38 | 0.97 [0.92-1.01] | 0.135 | 0.93 [0.86-1.01] |
| LSTD 3-month lag (°C) tercile | Tercile 1:26.36-<30.09 | Ref | | Ref |
| | Tercile 2:30.09-<32.96 | 0.99 [0.93-1.04] | 0.645 | 1.04 [0.98-1.10] |
| | Tercile 3:32.96-<41.38 | 1.05 [0.99-1.11] | 0.064 | **1.08 [1.00-1.17]** |

*(Continued)*

**Table 3.** (Continued)

| Variable | Category | Bivariate | | Multivariable |
|---|---|---|---|---|
| | | Unadjusted IRR [95% CI] | P-value | Adjusted IRR [95% CI] |
| Year | 2008 | Ref | | Ref |
| | 2009 | 0.92 [0.87-0.98] | 0.009 | 0.95 [0.88-1.02] |
| | 2010 | 0.78 [0.73-0.84] | **<0.001** | **0.77 [0.71-0.84]** |
| | 2011 | 0.80 [0.67-0.94] | 0.009 | 0.78 [0.61-1.00] |
| | 2012 | 0.81 [0.74-0.87] | **<0.001** | **0.87 [0.77-0.99]** |
| | 2015 | 0.71 [0.62-0.80] | **<0.001** | **0.82 [0.71-0.95]** |
| | 2016 | 0.74 [0.69-0.79] | **<0.001** | **0.82 [0.73-0.92]** |
| | 2017 | 0.76 [0.72-0.81] | **<0.001** | **0.82 [0.74-0.92]** |
| | 2018 | 0.72 [0.67-0.78] | **<0.001** | **0.77 [0.66-0.89]** |
| | 2022 | 0.65 [0.63-0.68] | **<0.001** | **0.77 [0.70-0.84]** |
| | 2023 | 0.70 [0.65-0.75] | **<0.001** | **0.80 [0.71-0.90]** |
| | 2024 | 0.65 [0.57-0.74] | **<0.001** | 0.87 [0.71-1.06] |

*IRR-Incidence Rate Ratios.

AFI-Acute Febrile illness; LSTD- Land Surface Temperature – Daytime.

β Cutoffs for respiratory rate based on the Pediatric Advanced Life Support (PALS) guidelines.

¥ Modified 17-point Vesikari score.

§ Full vaccinated- 1 dose of BCG, 3 doses of pentavalent and polio, and 1 dose of measles.

ŧ Unknown vaccination card not available.

reducing pediatric HIV infections and related immunosuppression, which increases vulnerability to severe diarrhea and other opportunistic infections [10]. This decline may have been further accentuated by the introduction and scaling up of Vitamin A supplementation through the Child Health Weeks campaign, launched in 2007. Increased coverage over time likely enhanced immune function, contributing to reduced severity and duration of diarrheal episodes and associated complications [33]. The observed reduction in comorbidities may also reflect advancements in local WASH infrastructure, a trend corroborated by the findings of our rotavirus vaccination sensitivity analysis. However, a paradox of this public health success combined with evolving diagnostic practices and underlying environmental factors is the emergence of a secondary layer of health issues that were previously overshadowed or misdiagnosed. Respiratory illness (non-pneumonia), AFI, and anemia have gained relative importance in the overall comorbidity burden of childhood diarrhea even as traditional comorbidities (malaria, pneumonia, wasting, and SAM) have declined.

Specifically, the introduction of the rotavirus vaccine in 2014 has reduced the incidence and severity of viral diarrhea [34,35]. The residual viral diarrheal episodes are likely less severe leading lower risk of severe dehydration, thereby reducing associated comorbidities. In addition, the observed reductions in comorbidity counts across all rotavirus vaccination categories in univariable analyses likely reflect herd immunity, whereby reduced population-level transmission confers indirect protection even among unvaccinated children [36]. However, we cannot rule out the contribution of broader public health improvements that affect all children regardless of their individual vaccination status. Furthermore, the updating of national guidelines for pediatric diarrhea to include zinc in 2007 alongside Oral rehydration solution (ORS) has worked to shorten duration and reduce severity of diarrhea, potentially lowering the risk of secondary complications [37]. In parallel, the introduction of the Pneumococcal Conjugate Vaccine (PCV10) into the national immunization program in 2011 has led to substantial reduction in PCV10-type pneumonia cases possibly explaining the reduced prevalence of pneumonia as a comorbidity [38]. This decline has been matched with an increase in respiratory illness (non-pneumonia), which likely reflects the persistent, and perhaps increasing, circulation of viral agents like Respiratory Syncytial Virus (RSV), influenza,

**Table 4. Factors associated with number of comorbidities in children aged 6-35 months with moderate-to-severe diarrhea in Western Kenya, 2008-2024.**

| Variable | Category | Bivariate | | Multivariable |
|---|---|---|---|---|
| | | Unadjusted IRR [95% CI] | P-value | Adjusted IRR [95% CI] |
| Age Categories | 6_11 m | Ref | | Ref |
| | 12_23 m | 1.02 [0.98-1.06] | 0.279 | 1.01 [0.96-1.07] |
| | 24_35 m | 1.05 [0.99-1.10] | 0.056 | **1.09 [1.02-1.15]** |
| Gender | Female | **0.93 [0.90-0.97]** | **0.001** | **0.94 [0.90-0.98]** |
| Caregiver education less than primary school | Yes | **1.18 [1.13-1.23]** | **<0.001** | **1.12 [1.07-1.17]** |
| Improved water | Unimproved | Ref | | Ref |
| | Improved | **0.93 [0.89-0.96]** | **<0.001** | **0.95 [0.91-0.99]** |
| Improved sanitation | Unimproved | Ref | | Ref |
| | Improved | **0.95 [0.91-0.98]** | **0.006** | 1.02 [0.97-1.07] |
| Respiratory Rate categories[ß] | Normal | Ref | | Ref |
| | Low | 0.91 [0.82-1.00] | 0.053 | 0.97 [0.86-1.10] |
| | High | **1.25 [1.19-1.32]** | **<0.001** | **1.18 [1.11-1.25]** |
| Max no. of watery diarrhea episodes in a day, n (%) | ≤ 6 | Ref | | Ref |
| | ≥7 | **0.91 [0.88-0.95]** | **<0.001** | 1.00 [0.95-1.06] |
| Pre-enrolment diarrhea days | | 1.01 [0.99-1.03] | 0.112 | 1.02 [0.99-1.04] |
| Experienced vomiting | | **1.23 [1.17-1.29]** | **<0.001** | **1.24 [1.17-1.31]** |
| Dehydration | None | Ref | | Ref |
| | Severe | **1.43 [1.27-1.61]** | **<0.001** | **1.36 [1.19-1.57]** |
| | Some | **1.23 [1.10-1.38]** | **<0.001** | **1.25 [1.10-1.43]** |
| Vesikari score Categories[¥] | Mild | Ref | | Ref |
| | Moderate | **1.29 [1.18-1.42]** | **<0.001** | – |
| | Severe | 1.54 [1.41-1.70] | **<0.001** | – |
| Fully vaccinated[§] | No | Ref | | Ref |
| | Yes | **0.95 [0.90-0.99]** | **0.030** | 0.99 [0.93-1.04] |
| | Unknown[ŧ] | **1.10 [1.05-1.16]** | **<0.001** | 1.08 [0.99-1.16] |
| PCV10 vaccine | Unvaccinated | Ref | | Ref |
| | Vaccinated (≥ 1) | **0.86 [0.75-0.98]** | **0.022** | **0.84 [0.73-0.97]** |
| | Unknown[ŧ] | 1.04 [0.91-1.19] | 0.585 | 0.87 [0.74-1.02] |
| Rainfall (mm) tercile | Tercile 1:13.92-<82.49 | Ref | | Ref |
| | Tercile 2:82.49-<158.30 | 1.00 [0.95-1.06] | 0.860 | 1.04 [0.96-1.13] |
| | Tercile 3:158.30-<353.40 | 0.99 [0.94-1.04] | 0.683 | 1.04 [0.95-1.13] |
| Rainfall 3-month lag (mm) tercile | Tercile 1:13.92-<82.49 | Ref | | Ref |
| | Tercile 2:82.49-<158.30 | 1.05 [0.99-1.11] | 0.101 | 1.05 [0.98-1.13] |
| | Tercile 3:158.30-<353.40 | 1.02 [0.96-1.09] | 0.462 | 1.06 [0.98-1.14] |
| LSTD (°C) tercile | Tercile 1:26.36-<30.09 | Ref | | Ref |
| | Tercile 2:30.09-<32.96 | 0.98 [0.93-1.03] | 0.368 | 1.00 [0.94-1.06] |
| | Tercile 3:32.96-<41.38 | 0.98 [0.93-1.03] | 0.519 | 0.94 [0.85-1.03] |
| LSTD 3-month lag (°C) tercile | Tercile 1:26.36-<30.09 | Ref | | Ref |
| | Tercile 2:30.09-<32.96 | 1.03 [0.96-1.09] | 0.414 | **1.07 [1.00-1.14]** |
| | Tercile 3:32.96-<41.38 | **1.08 [1.01-1.14]** | **0.018** | **1.10 [1.01-1.20]** |

*(Continued)*

**Table 4.** (Continued)

| Variable | Category | Bivariate | | Multivariable |
|---|---|---|---|---|
| | | Unadjusted IRR [95% CI] | P-value | Adjusted IRR [95% CI] |
| Year | 2008 | Ref | | Ref |
| | 2009 | **0.90 [0.84-0.96]** | **0.002** | 0.94 [0.86-1.02] |
| | 2010 | **0.78 [0.72-0.85]** | **<0.001** | **0.78 [0.71-0.86]** |
| | 2011 | 0.86 [0.71-1.04] | 0.119 | 0.85 [0.64-1.14] |
| | 2012 | **0.79 [0.72-0.87]** | **<0.001** | **0.85 [0.74-0.97]** |
| | 2015 | **0.74 [0.64-0.85]** | **<0.001** | 0.87 [0.73-1.03] |
| | 2016 | **0.73 [0.67-0.79]** | **<0.001** | **0.79 [0.69-0.90]** |
| | 2017 | **0.75 [0.71-0.81]** | **<0.001** | **0.83 [0.74-0.94]** |
| | 2018 | **0.73 [0.67-0.79]** | **<0.001** | **0.74 [0.63-0.88]** |
| | 2022 | **0.63 [0.60-0.66]** | **<0.001** | **0.76 [0.69-0.85]** |
| | 2023 | **0.67 [0.63-0.72]** | **<0.001** | **0.79 [0.70-0.89]** |
| | 2024 | **0.63 [0.55-0.72]** | **<0.001** | 0.87 [0.71-1.06] |

*IRR-Incidence Rate Ratios.

AFI-Acute Febrile illness; LSTD- Land Surface Temperature – Daytime.

β Cutoffs for respiratory rate based on the Pediatric Advanced Life Support (PALS) guidelines.

¥ Modified 17-point Vesikari score.

§ Full vaccinated- 1 dose of BCG, 3 doses of pentavalent and polio, and 1 dose of measles.

ŧ Unknown vaccination card not available.

parainfluenza, and more recently, SARS-CoV-2 [39,40]. Additionally, urbanization, increased population density, and environmental changes including climate variability may be facilitating the spread of respiratory viruses [41]. The distinct peaks in respiratory illness observed in 2022–2023 may also reflect post-COVID-19 lockdown surges in common viral pathogens as population immunity shifted. Additionally,

revisions to IMCI guidelines in 2014 that refined pneumonia classification and emphasized objective assessment of hypoxaemia using pulse oximetry, may have led to reclassification of milder respiratory presentations [42]. Similarly, gradual reductions in childhood stunting have also been reported in the country, with a reduction rate of 1.6% per annum reported between 1993–2014 largely driven by improved national socio-economic well-being [43]. The most recent 2022 Kenya Demographic Health Survey reported a prevalence of 18.0% [11].

In synchrony, the significant drop in malaria as a comorbidity directly correlates with the nationwide scale-up of insecticide-treated nets (ITNs) and effective artemisinin-based combination therapy (ACT), which began in 2006 [44]. This reduction in malaria prevalence has also been heightened by the pilot introduction of the RTS,S malaria vaccine in high-burden counties including Siaya County in 2019 [45]. Additionally, the widespread use of Rapid Diagnostic Tests (RDTs) and microscopy has reduced the over-diagnosis of malaria [46], directly explaining the rise in AFI, which is often a diagnosis of exclusion, capturing a wide range of viral (including dengue, chikunguya) and bacterial infections (salmonellosis, rickettsia). Of note, primary healthcare settings lack the capacity to diagnose the etiology of AFI beyond malaria testing, leading to this broad category gaining prominence. Additionally, the emergence of anemia is consistent with findings from Hailu et al., which showed an increase in the prevalence of anemia in the East Africa region in the period 2016–2021 following gradual declines in 2006–2015 [47]. This could possibly be explained by the fact that despite the decline in malaria, malaria remains endemic in the study area possibly causing hemolytic anemia in young children [21]. Additionally, anemia is closely linked to poor nutrition, with stunted and wasted children more affected than their peers possibly caused by inadequate maternal nutrition, and low intake of iron-rich foods [48].The impact of the interventions and vaccinations

programs discussed above may explain in part the decline in prevalence of the leading comorbidities as well as the overall burden of comorbidity count in childhood diarrhea.

The higher comorbidity count in older children compared to younger children likely reflects increased environmental exposure due to greater mobility and social interaction [49], combined with a maturing immune system that is still developing after the loss of maternal antibodies [50], making them more vulnerable to overlapping infections. Moreover, lower caregiver education is associated with higher comorbidity in children likely due to limited health knowledge, delayed care-seeking, poor hygiene and nutrition practices, and overall socioeconomic disadvantage [51]. These factors increase children's exposure to infections and reduce their chances of timely treatment, leading to a higher risk of multiple, overlapping illnesses. Dehydration, vomiting, and high respiratory rate are predictors of higher comorbidity count because they signal severe physiological disruption, creating a cascade of vulnerability. Dehydration strains body organs and disrupts metabolism, leading to acidosis and tissue hypoxia, which weakens defenses and enables secondary infections [52]. Vomiting prevents rehydration and nutrition, causing electrolyte imbalances, catabolism, and immunosuppression. Similarly, a high respiratory rate serves as a dual indicator, pointing either directly to a pulmonary comorbidity or to the body's attempt to compensate for metabolic acidosis driven by the combined severity of multiple illnesses [53]. Collectively, these signs signify a cycle of organ dysfunction and immune compromise that elevates susceptibility to secondary infections, thereby increasing the total comorbidity burden.

The higher comorbidity count among children with unknown vaccination status likely reflects the role of the immunization card as a proxy for broader socio-economic vulnerability [54,55]. Unavailability of health records is likely associated with household instability, lower health literacy, and poorer access to improved water and sanitation as has been documented in a previous study in Malaysia [54]. These factors converge to increase the environmental pathogen burden, making polymicrobial infections more likely among children from households struggling to maintain formal health documentation. The absence of immunization cards could also be a proxy for lower engagement with preventive health services, further contributing to higher disease burden in this group. The association between 3-month lagged environmental variables and increased comorbidity counts (aIRR 1.07–1.08) suggests an ecological priming effect [56,57]. High rainfall and temperatures may facilitate the gradual accumulation of enteric pathogens in environmental reservoirs or trigger shifts in nutritional security and vector density. These delayed effects likely increase the environmental pathogen pressure, making polymicrobial exposure more probable for vulnerable children three months following climatic extremes.

Conversely, females were more likely to have fewer comorbidities, a finding that is corroborated by Muenchhoff et al., who observed a strong sexual dimorphism in childhood infection outcomes, with males often more susceptible to many pathogens [58]. This may be partly due to females exhibiting stronger Th1 immune responses, even before puberty and full expression of sexual traits.

Our findings highlight the critical need for sustained investment in interventions targeting traditional comorbidities including vaccination, malaria control, PMTCT of HIV, and nutrition initiatives to maintain the successes witnessed. However, there is need for public health priorities to pivot to address emerging comorbidities (respiratory illness [non-pneumonia], AFI, and anemia) by tackling the multifaceted drivers of anemia, respiratory risk factors, and improving diagnostics and management for AFI. Moreover, the frequent co-occurrence of stunting with infectious diseases demonstrates that siloed programs are insufficient. Our findings advocate for integrated care packages that combine nutrition support, vaccination, WASH promotion, and infectious disease management at every point of contact with the health system.

Our study is not without limitations. First, reliance on clinician diagnoses for comorbidities introduces potential subjectivity and misclassification bias. Second, the temporal sequence of co-occurring conditions is unclear: whether they developed concurrently or sequentially. Third, treating all comorbidities equally overlooks clinical and temporal differences between chronic and acute conditions. Fourth, restricting the analysis to one Kenyan site may limit generalizability of findings to other settings with different epidemiological or socio-economic profiles. The 16-year timeframe also introduces potential for unmeasured temporal confounders such as evolving healthcare access or public health interventions. Lastly,

despite covariate adjustment, unmeasured factors like care-seeking behavior or underlying immunodeficiency may still confound associations.

## Conclusion

Over the past 16 years, Kenya has achieved a remarkable epidemiological transition in childhood diarrheal morbidity, marked by a 20–23% decline in comorbidity burden, despite the high prevalence, and a fundamental shift in disease profiles. The reduction in severe traditional comorbidities (malaria, pneumonia, wasting, and stunting) stands as a testament to the success of scaled interventions. However, this progress has unmasked a new challenge: the rising relative burden of anemia, respiratory illness (non-pneumonia), and AFI, now dominant features of the comorbidity landscape. Our findings serve as a call to action for Kenya's strategy to evolve from single-disease control to integrated comorbidity management.

## Supporting information

**S1 Table. Diarrheal comorbidities based on Integrated Management of Childhood Illness (IMCI) case definitions and clinician diagnosis.**
(DOCX)

**S2 Table. Factors associated with the number of comorbidities among children aged 0–59 months presenting with moderate-to-severe diarrhea in Western Kenya, 2008–2024, including rotavirus vaccination status.**
(DOCX)

## Acknowledgments

We extend our deep gratitude to the families who participated in the three studies, and to the clinical and field teams whose dedication and hard work made this research possible. We are also grateful to the physicians, administrators, and Ministry of Health officials at each recruitment center in Siaya County for their provision of facilities, operational support, and gatekeeper permissions that enabled the successful implementation of the studies.

The findings and conclusions in this report are those of the authors and do not necessarily represent the official position of the Kenya Medical Research Institute or partnering institutions.

## Author contributions

**Conceptualization:** Billy Ogwel, Elizabeth T Rogawski McQuade.

**Data curation:** Billy Ogwel, Bryan O. Nyawanda.

**Formal analysis:** Billy Ogwel, Bryan O. Nyawanda.

**Funding acquisition:** Karen L. Kotloff, Patricia B. Pavlinac, Richard Omore.

**Investigation:** Dilruba Nasrin, Karen L. Kotloff, Patricia B. Pavlinac, Richard Omore, Elizabeth T Rogawski McQuade.

**Methodology:** Billy Ogwel, Alex O. Awuor, Raphael O. Anyango, Karen L. Kotloff, Patricia B. Pavlinac, Richard Omore, Elizabeth T Rogawski McQuade.

**Supervision:** Elizabeth T Rogawski McQuade.

**Visualization:** Billy Ogwel.

**Writing – original draft:** Billy Ogwel.

**Writing – review & editing:** Bryan O. Nyawanda, Brian O. Onyando, Alex O. Awuor, Caleb Okonji, Raphael O. Anyango, Caren Oreso, Catherine Sonye, John B. Ochieng, Stephen Munga, Dilruba Nasrin, Karen L. Kotloff, Patricia B. Pavlinac, Richard Omore, Elizabeth T Rogawski McQuade.

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
