## [Decision Letter · Decision Letter 0]

6 Jan 2026

PGPH-D-25-03455

Temporal Patterns and Risk Factors of Diarrheal Comorbidity among Children aged < 5 years in Rural Western Kenya: Evidence from Three Consecutive Enteric Studies─2008-2024

Dear <!--StartFragmentMr Billy Ogwel<!--EndFragment

Thank you for submitting your manuscript to PLOS Global Public Health. After careful consideration, we feel that it has merit but does not fully meet PLOS Global Public Health’s publication criteria as it currently stands. Therefore, we invite you to submit a revised version of the manuscript that addresses the points raised during the review process.

The manuscript has been evaluated by external reviewers, whose detailed comments are provided below. Both reviewers acknowledge the importance of the topic and note that the manuscript is generally well written and well organized. The pooled analysis of data from GEMS, VIDA, and EFGH is considered a valuable contribution with relevance for diarrheal disease control and management in children under five years of age. However, several substantive concerns have been raised that must be addressed before the manuscript can be considered further.<o:p></o:p>*<o:p></o:p>*

We look forward to receiving your revised manuscript.

Kind regards,

Tintu Varghese, MD

Academic Editor

Journal Requirements:

Please send a completed 'Competing Interests' statement, including any COIs declared by your co-authors. If you have no competing interests to declare, please state "The authors have declared that no competing interests exist".

Please amend your detailed Financial Disclosure statement. This is published with the article. It must therefore be completed in full sentences and contain the exact wording you wish to be published.a. State the initials, alongside each funding source, of each author to receive each grant.

We have noticed that you have uploaded Supporting Information files, but you have not included a list of legends. Please add a full list of legends for your Supporting Information files after the references list.

Additional Editor Comments (if provided):

Thank you for submitting your manuscript entitled “Temporal Patterns and Risk Factors of Diarrheal Comorbidity among Children aged < 5 years in Rural Western Kenya: Evidence from Three Consecutive Enteric Studies─2008-2024”to PLOS Global Public Health.

The manuscript has been evaluated by external reviewers, whose detailed comments are provided below. Both reviewers acknowledge the importance of the topic and note that the manuscript is generally well written and well organized. The pooled analysis of data from GEMS, VIDA, and EFGH is considered a valuable contribution with relevance for diarrheal disease control and management in children under five years of age. However, several substantive concerns have been raised that must be addressed before the manuscript can be considered further.

Reviewer 1

P124: Both GEMS and VIDA enrolled children under 5 while EFGH enrolled children aged 6-35 months. Please clarify whether this difference in age distribution or missing 0-5 months and 36-59 months would affect the analysis, especially growth related factors.

P137: Please clarify whether “bacterial infection” excluded diarrhea associated bacterial infection?

P138: Would add an Oxford comma after “stunting”.

Figure 4: I would suggest to remove the numbers at the bottom.

Table S1: As comorbidities were the focus of the this work, I would suggest to add more details for “other clinical symptoms” regarding case definitions.

Reviewer 2

I don't see a clear statement on whether rotavirus vaccination was assessed in the analyses. It is plausible that rotavirus vaccination may impact the risk of the comorbidities examined here, in particular severe acute malnutrition. Please provide some further information regarding how that was handled. Please also indicate if vaccine status (receipt of core vaccines, being fully vaccinated) was assessed in the analyses? I think some discussion of the impact of a major vaccine against diarrheal disease is warranted when assessing the risk factors for diarrheal comorbidities.

I'm missing the utility of the climate data that is presented here. Very little reference is made in the discussion section to the possible role it may have as risk factor for diarrheal comorbidities. While climate metrics are important risk factors for disease burden shifts, they were not clearly identified as a risk factor here and the authors may wish to reconsider including it in this paper.

Introduction:

Lines 82-24: Consider adding a sentence regarding the wider availability and use of Oral Rehydration Solution (ORS) as a relevant public health intervention

Methods:

Please include in lines 100-104 when Kenya introduced rotavirus vaccine and coverage rates.

Did you collect data on vaccination status? Do you have information on rotavirus vaccination status for those children eligible to receive it in VIDA or EFGH?

Can you please clarify at what level did you collect climate covariates? Were these collected for all of Western Kenya, for Siaya County or for each individual participant’s address?

Results:

Line 250, please provide the peaks of AFI you are referring to from Table 2 in the text.

Table 3 the rain and LSTD variables should be more clearly specified so that the table stands independently. I assume that Lag 1 is one month lag, 2 month lag and 3 month lag respectively but this could be made clearer in the table. This table is also rather cumbersome and it may benefit from reducing the rain and LSTD variables to present only the LSTD 3 month lag category.

Discussion

The authors provide valid explanations for the impact rotavirus vaccine and PCV10 have had on the reduction of viral diarrhea and on pneumonia cases respectively. They also provide a good explanation in lines 335-337 on the persistent and plausibly increasing circulation of other viral respiratory pathogens. Is it possible that some of the observed increase in respiratory illness comorbidities may also be due to evolving diagnostic options and awareness in the region since the 2009 influenza pandemic and COVID pandemic. Were changes made to ICMI criteria to expand diagnoses of non-pneumonia respiratory illness during the time frame of their study?

In light of these comments, the editorial decision is Major Revision. We invite you to revise the manuscript by carefully addressing all reviewer comments and providing a detailed, point-by-point response explaining how each concern has been handled. Where suggested analyses or data are not available, please clearly state this and discuss the implications as limitations.

Please note that submission of a revised manuscript does not guarantee acceptance, but we believe that a thorough revision has the potential to substantially strengthen the paper.

We look forward to receiving your revised manuscript and thank you for considering PLOS Global Public Health for your work.

Reviewers' comments:

Reviewer's Responses to Questions

**Comments to the Author**

1. Does this manuscript meet PLOS Global Public Health’s publication criteria?

Reviewer #1: Yes

Reviewer #2: Yes

2. Has the statistical analysis been performed appropriately and rigorously?

Reviewer #1: Yes

Reviewer #2: Yes

3. Have the authors made all data underlying the findings in their manuscript fully available (please refer to the Data Availability Statement at the start of the manuscript PDF file)?

Reviewer #1: Yes

Reviewer #2: Yes

4. Is the manuscript presented in an intelligible fashion and written in standard English?

Reviewer #1: Yes

Reviewer #2: Yes

Reviewer #1: The manuscript analyzed temporal patterns and risk factors of diarrheal comorbidity in Kenyan children aged < 5, leveraging three important diarrhea etiology/surveillance studies, i.e. GEMS, VIDA, and EFGH. The findings revealed a decline trend in comorbidity burden and a shift in disease profiles, providing evidences for disease control and management. It was well-written and organized.

P124: Both GEMS and VIDA enrolled children under 5 while EFGH enrolled children aged 6-35 months. Please clarify whether this difference in age distribution or missing 0-5 months and 36-59 months would affect the analysis, especially growth related factors.

P137: Please clarify whether “bacterial infection” excluded diarrhea associated bacterial infection?

P138: Would add an Oxford comma after “stunting”.

Figure 4: I would suggest to remove the numbers at the bottom.

Table S1: As comorbidities were the focus of the this work, I would suggest to add more details for “other clinical symptoms” regarding case definitions.

Reviewer #2: This paper presents an analysis of the temporal patterns and risk factors for diarrheal comorbidities in Kenyan children. They conducted a secondary pooled analysis using a retrospective cohort design to leverage 3 well established diarrheal studies. The topic is of importance and interest, as the epidemiology of diarrheal diseases continues to evolve and a better understanding of the public health implications of disease shifts is needed. Overall, the paper is well written and clearly describes the study and the results obtained.

Major comments:

I don't see a clear statement on whether rotavirus vaccination was assessed in the analyses. It is plausible that rotavirus vaccination may impact the risk of the comorbidities examined here, in particular severe acute malnutrition. Please provide some further information regarding how that was handled. Please also indicate if vaccine status (receipt of core vaccines, being fully vaccinated) was assessed in the analyses? I think some discussion of the impact of a major vaccine against diarrheal disease is warranted when assessing the risk factors for diarrheal comorbidities.

I'm missing the utility of the climate data that is presented here. Very little reference is made in the discussion section to the possible role it may have as risk factor for diarrheal comorbidities. While climate metrics are important risk factors for disease burden shifts, they were not clearly identified as a risk factor here and the authors may wish to reconsider including it in this paper.

Minor comments:

Introduction:

Lines 82-24: Consider adding a sentence regarding the wider availability and use of Oral Rehydration Solution (ORS) as a relevant public health intervention

Methods:

Please include in lines 100-104 when Kenya introduced rotavirus vaccine and coverage rates.

Did you collect data on vaccination status? Do you have information on rotavirus vaccination status for those children eligible to receive it in VIDA or EFGH?

Can you please clarify at what level did you collect climate covariates? Were these collected for all of Western Kenya, for Siaya County or for each individual participant’s address?

Results:

Line 250, please provide the peaks of AFI you are referring to from Table 2 in the text.

Table 3 the rain and LSTD variables should be more clearly specified so that the table stands independently. I assume that Lag 1 is one month lag, 2 month lag and 3 month lag respectively but this could be made clearer in the table. This table is also rather cumbersome and it may benefit from reducing the rain and LSTD variables to present only the LSTD 3 month lag category.

Discussion

The authors provide valid explanations for the impact rotavirus vaccine and PCV10 have had on the reduction of viral diarrhea and on pneumonia cases respectively. They also provide a good explanation in lines 335-337 on the persistent and plausibly increasing circulation of other viral respiratory pathogens. Is it possible that some of the observed increase in respiratory illness comorbidities may also be due to evolving diagnostic options and awareness in the region since the 2009 influenza pandemic and COVID pandemic. Were changes made to ICMI criteria to expand diagnoses of non-pneumonia respiratory illness during the time frame of their study?

**Do you want your identity to be public for this peer review?** For information about this choice, including consent withdrawal, please see our Privacy Policy

Reviewer #1: No

Reviewer #2: No

---

## [Editor Report · Decision Letter 1]

26 Jan 2026

Temporal Patterns and Risk Factors of Diarrheal Comorbidity among Children aged < 5 years in Rural Western Kenya: Evidence from Three Consecutive Enteric Studies─2008-2024

PGPH-D-25-03455R1

Dear Dr. Billy Ogwel

We are pleased to inform you that your manuscript 'Temporal Patterns and Risk Factors of Diarrheal Comorbidity among Children aged < 5 years in Rural Western Kenya: Evidence from Three Consecutive Enteric Studies─2008-2024' has been provisionally accepted for publication in PLOS Global Public Health.

You have addressed the reviewers comments and revised manuscript to strengthen it further. The study’s objectives, design, and key findings are clearly presented, and the analysis across GEMS, VIDA, and EFGH datasets provides important insights into temporal trends and risk factors for diarrheal comorbidities in Kenyan children. Your findings on the declining comorbidity burden and shifting disease profiles offer valuable evidence to inform integrated disease management and public health strategies in sub-Saharan Africa.

Best regards,

Tintu Varghese, MD

Academic Editor
